# Piezo1 is a mechanically activated ion channel and mediates pressure induced pancreatitis

Joelle M.-J. Romac[1], Rafiq A. Shahid[1], Sandip M. Swain[1], Steven R. Vigna[1,2] & Rodger A. Liddle[1]

Merely touching the pancreas can lead to premature zymogen activation and pancreatitis but the mechanism is not completely understood. Here we demonstrate that pancreatic acinar cells express the mechanoreceptor Piezo1 and application of pressure within the gland produces pancreatitis. To determine if this effect is through Piezo1 activation, we induce pancreatitis by intrapancreatic duct instillation of the Piezo1 agonist Yoda1. Pancreatitis induced by pressure within the gland is prevented by a Piezo1 antagonist. In pancreatic acinar cells, Yoda1 stimulates calcium influx and induces calcium-dependent pancreatic injury. Finally, selective acinar cell-specific genetic deletion of Piezo1 protects mice against pressure-induced pancreatitis. Thus, activation of Piezo1 in pancreatic acinar cells is a mechanism for pancreatitis and may explain why pancreatitis develops following pressure on the gland as in abdominal trauma, pancreatic duct obstruction, pancreatography, or pancreatic surgery. Piezo1 blockade may prevent pancreatitis when manipulation of the gland is anticipated.

[1] Department of Medicine, Duke University and Durham VA Medical Centers, Durham, NC 27710, USA. [2] Department of Cell Biology, Duke University and Durham VA Medical Centers, Durham, NC 27710, USA. These authors contributed equally: Joelle M.-J. Romac, Rafiq A. Shahid. Correspondence and requests for materials should be addressed to R.A.L. (email: rodger.liddle@duke.edu)

The pancreas is unusually sensitive to mechanical injury. It has long been recognized that manipulation of the pancreas at the time of surgery can induce acute pancreatitis complicating postoperative recovery. The condition is so well-known among surgeons that they avoid manipulating the pancreas whenever possible. In addition, blunt trauma to the abdomen that occurs following blows to the abdomen or in motor vehicle accidents is an all too common cause of pancreatitis[1]. For nearly 100 years it was believed that gallstone impaction at the junction of the bile and pancreatic ducts produced pancreatitis by promoting the reflux of bile into the pancreas[2]. However, later studies indicated that backpressure induced by occlusion of the pancreatic duct could cause pancreatitis[3]. Thus, increased pressure within the gland could be responsible for gallstone pancreatitis. This concept has been reinforced by the clinical observation that overfilling the pancreatic duct during endoscopic retrograde cholangiopancreatography (ERCP), a diagnostic radiographic tool to visualize the pancreas, can trigger acute pancreatitis[4]. Although the mechanism responsible for initiating the injury during ERCP is not completely understood, it is believed that increased intraductal pressure could be responsible[5]. It appears, therefore, that conditions that produce pressure on the gland can cause pancreatitis suggesting that the pancreas itself can perceive mechanical force.

The recent discovery of a novel class of pressure-activated ion channels, Piezo1 and Piezo2, led us to consider whether mechanically activated ion channels exist in the pancreas. Piezo1 and Piezo2 are the two members of a distinct nonselective cationic mechanosensitive channel family expressed in mammalian cells (for review see Gottlieb and Sachs[6]). Piezo1 was first identified in a neuronal cell line using mechanical stimulation[7]. The Piezo2 protein was found subsequently through sequence homology[7]. The Piezo proteins are present in numerous mammalian tissues with particularly high expression in lung, bladder, and skin. Piezo1 protein is comprised of roughly 2500 amino acids. Recent structure data obtained by cryo-electron microscopy revealed that the mouse Piezo1 has at least 26 transmembrane domains[8] and up to 38 transmembrane domains based on varied algorithms[9,10]. Piezo1 can form a homotrimer with an extracellular domain consisting of a propeller-like structure believed to act as a force sensor and thought to be involved in gating the ion-conducting pore[11]. Purified Piezo1 incorporated into lipid bilayers inherently responds to mechanical forces[12] and allows the flow of cations across the membrane. The Piezo1 channel exhibits a preference for calcium in response to stimulation in whole-cell recording or in outside-out patches[7,13]. Piezo1 responds to static pressure, shear stress (fluid flow), and membrane stretch[7,14,15]. Activation of Piezo1 can be blocked by GsMTx4, a peptide isolated from the tarantula spider *Grammostola spatulata*, which appears to act as a gating modifier since increased pressure can overcome its inhibitory effect[16]. Piezo1, but not Piezo2, activation can be evoked in the absence of mechanical stretch by the small molecule Yoda1[12].

Piezo1 plays a physiological role in several tissues, including vascular endothelium where it responds to the shear stress of blood flow. Deletion of the *Piezo1* gene is lethal during early embryogenesis[17] and vascular malformations result from targeted deletion of the *Piezo1* gene in the endothelium within days of the heart beating[15]. Piezo1 is also highly expressed in the urinary bladder where it responds to mechanical stretch[18]. A mechanism for mechanosensation in the pancreas has not been identified; however, we postulated that a mechanically activated ion channel might be involved in the deleterious effects of pressure causing pancreatitis.

By virtue of its uniquely robust production of digestive enzymes, perturbations of the pancreatic acinar cell initiate a series of events leading to pancreatitis[19]. Intracellular calcium concentrations are precisely regulated within the acinar cell. Abnormally high intracellular calcium concentrations can cause trypsin activation and impaired zymogen granule processing[20,21]. Fusion of zymogen granules with lysosomes causes activation of trypsinogen and other pancreatic proenzymes, which promote autodigestion of the pancreas[22]. An accompanying inflammatory reaction produces local and, if uncontrolled, deleterious systemic complications[23]. Thus, precise regulation of intracellular calcium concentrations is important for normal acinar cell homeostasis.

In the studies described below, we demonstrate that the cation channel Piezo1 is expressed in pancreatic acinar cells and pathological Piezo1 activation initiates pressure-induced acute pancreatitis. Furthermore, blockade of Piezo1 channels and specific pancreatic acinar cell Piezo1 deletion protects mice from pancreatitis.

## Results

**Intraductal pressure induces acute pancreatitis.** In order to produce a clinically relevant model for applying pressure to the pancreas we infused buffered saline solution (pH 7.6) into the pancreatic duct (Fig. 1a). Intraductal application of pressure avoids complications of tissue crush injury that we observed when attempting to apply pressure to the pancreas via external sources (e.g., pinching or applying weight to the gland). Fluid infused at 5 µL/min for 10 min produced a pressure of 7–11 mm Hg; a condition that we called low-pressure input. Infusion of saline at a rate of 80 µL/min for 5 min produced a pressure of 25–33 mm Hg that we refer to as high pressure.

Methylene blue (1%) added to the saline solution demonstrated that the injectate extended from the head of the pancreas into the pancreatic tail (Fig. 1b). Based on these observations we harvested the mid-portion of the pancreas that corresponds to the gastric and proximal duodenal lobes[24] for biochemical and histologic evaluation 24 h post duct injection. As shown in Fig. 1, high-pressure injections caused significant increases in pancreatic weight (Fig. 1c) and significant elevations in serum amylase levels (Fig. 1d). Pancreatic myeloperoxidase (MPO) concentrations (Fig. 1e), indicative of neutrophil infiltration, were also significantly elevated and histological changes demonstrated neutrophil infiltration, edema, hemorrhage, and tissue necrosis (Fig. 1f). These findings demonstrate that increasing intraductal pressure within the pancreas causes acute pancreatitis.

**Piezo1 is expressed in pancreatic acinar cells.** To determine if the effects of pressure on the pancreas could be induced by mechanically activated ion channels we examined pancreatic acinar cells for expression of channels previously demonstrated to respond to stretch or pressure. By quantitative reverse transcription-PCR (RT-PCR) we analyzed the expression of members of the TRPC[25], the DEG/ENaC[26], and the Piezo[27] families in pancreatic acinar cells prepared from C57BL/6J mice. The messenger RNA for *Piezo1* was the most abundant of the genes tested (Fig. 2a). Visualization by confocal microscopy of frozen sections from murine pancreatic tissue illustrated the presence of the Piezo1 protein in acinar cells (Fig. 2b and Supplementary Fig. 1). Piezo1 appears prominently on the surface of acinar cells, which were identified by their characteristic shape within the acinus and the presence of trypsin staining in zymogen granules toward the apical portion of the cells.

**Regulation of intracellular calcium.** To determine if Piezo1 activation influences intracellular calcium in pancreatic acinar cells we examined the effects of the Piezo1 agonist Yoda1. As shown in Fig. 2c, d and Supplementary Movie 1, Yoda1 increased

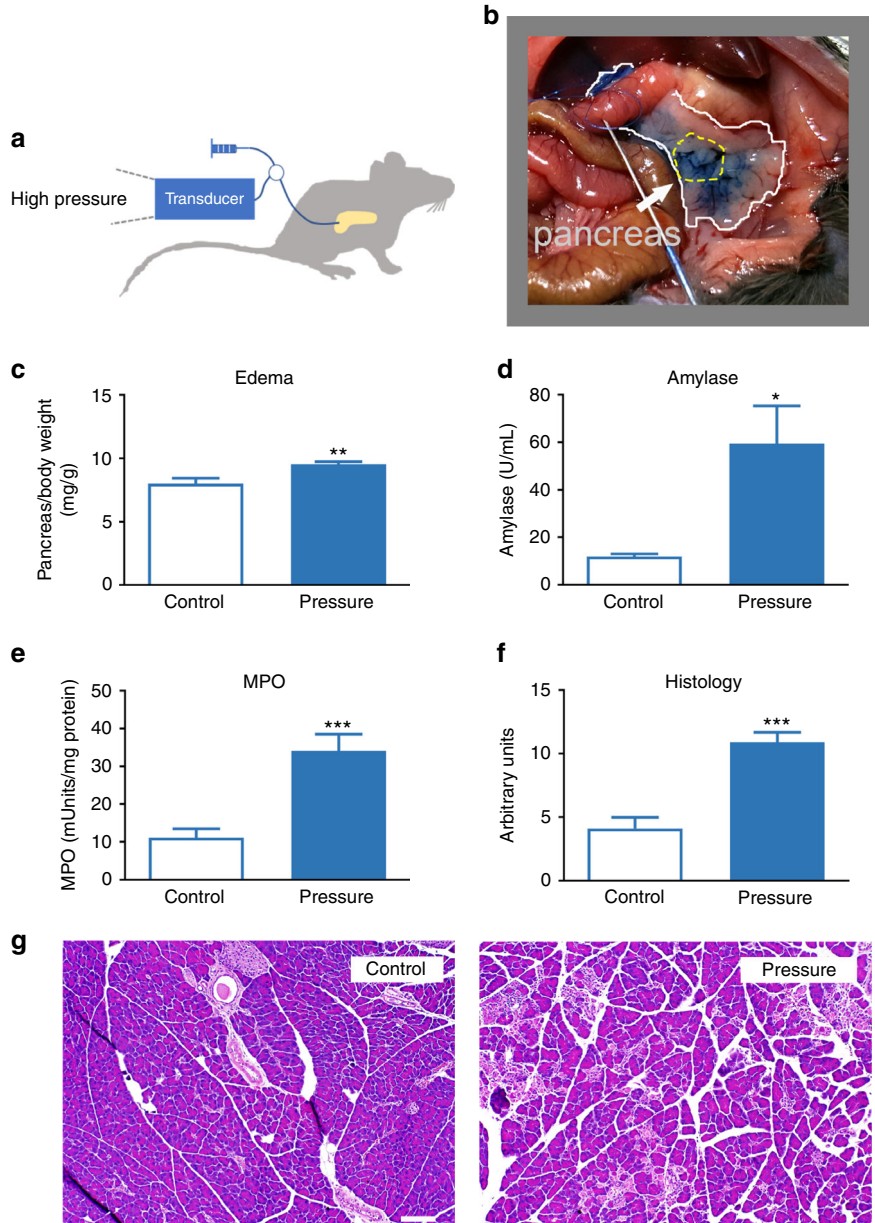

**Fig. 1** Intrapancreatic duct pressure causes acute pancreatitis. **a** Test solutions were infused into the pancreatic duct of mice as illustrated. Injection pressures were continuously monitored using a pressure transducer. **b** Injection of 400 μL buffered saline containing methylene blue dye was distributed throughout the pancreas. The white line indicates the boundary of the pancreas and yellow line indicates the mid-region of the pancreas that was used for histology and myeloperoxidase (MPO) determinations. Either sham-operated animals (control) or buffered saline solutions were infused at 25–33 mm Hg (pressure) and **c** edema, **d** blood amylase, **e** MPO, and **f** histological scoring were analyzed 24 h after duct injection in C57BL/6J male mice. Statistical analysis was performed using Student's $t$ test. $*P \leq 0.05$; $**P \leq 0.01$; $***P \leq 0.001$ ($n = 7$–13 for control and 13–17 for pressure). **g** Representative images of pancreatic tissue stained with H&E. Bar = 100 μm

$[Ca^{2+}]_i$ in pancreatic acinar cells loaded with the calcium indicator Calcium 6-QF. The peptide GsMTx4 is the only antagonist known to block mechanosensitive cation channels[28]. More precisely, GsMTx4 has been shown to inhibit Piezo1 overexpressed in HEK293 cells[16]. Importantly in acinar cells, the increase in $[Ca^{2+}]_i$ induced by Yoda1 was inhibited by GsMTx4 (Fig. 2d, e and Supplementary Movie 2).

In order to confirm the specificity of these effects we developed a mouse line that, through Cre expression driven by the *Ptf1a* promoter, ablated Piezo1 exclusively in pancreatic acinar cells. Cre expression was induced by tamoxifen treatments that were performed on mice 4 weeks of age. At this age, Ptf1a is expressed

only in pancreatic acinar cells[29]. The penetrance of Ptf1a-Cre expression in acinar cells was assessed by visualizing fluorescence emitted by enhanced yellow fluorescence protein (EYFP) in acini of mice carrying the genotype *Ptf1a-Cre^{ER};Gt(ROSA)26Sor^{tm3 (CAG-EFYP)};Piezo1^{fl/fl}*. These mice express EYFP when the floxed sequences encompassing the stop codon CAG are removed. After tamoxifen induction over 93% of pancreatic acinar cells from these mice expressed the Cre recombinase (Supplementary Fig. 2) suggesting that mice bearing the *Ptf1a-Cre^{ER};Piezo1^{fl/fl}* genotype largely lack a functional Piezo1 protein in pancreatic acinar cells. (See Methods for generation of *Ptf1a^{CreERT};Piezo1^{fl/fl}* mice, which we refer to as Piezo1^{aci} KO mice.) Unlike their wild-type

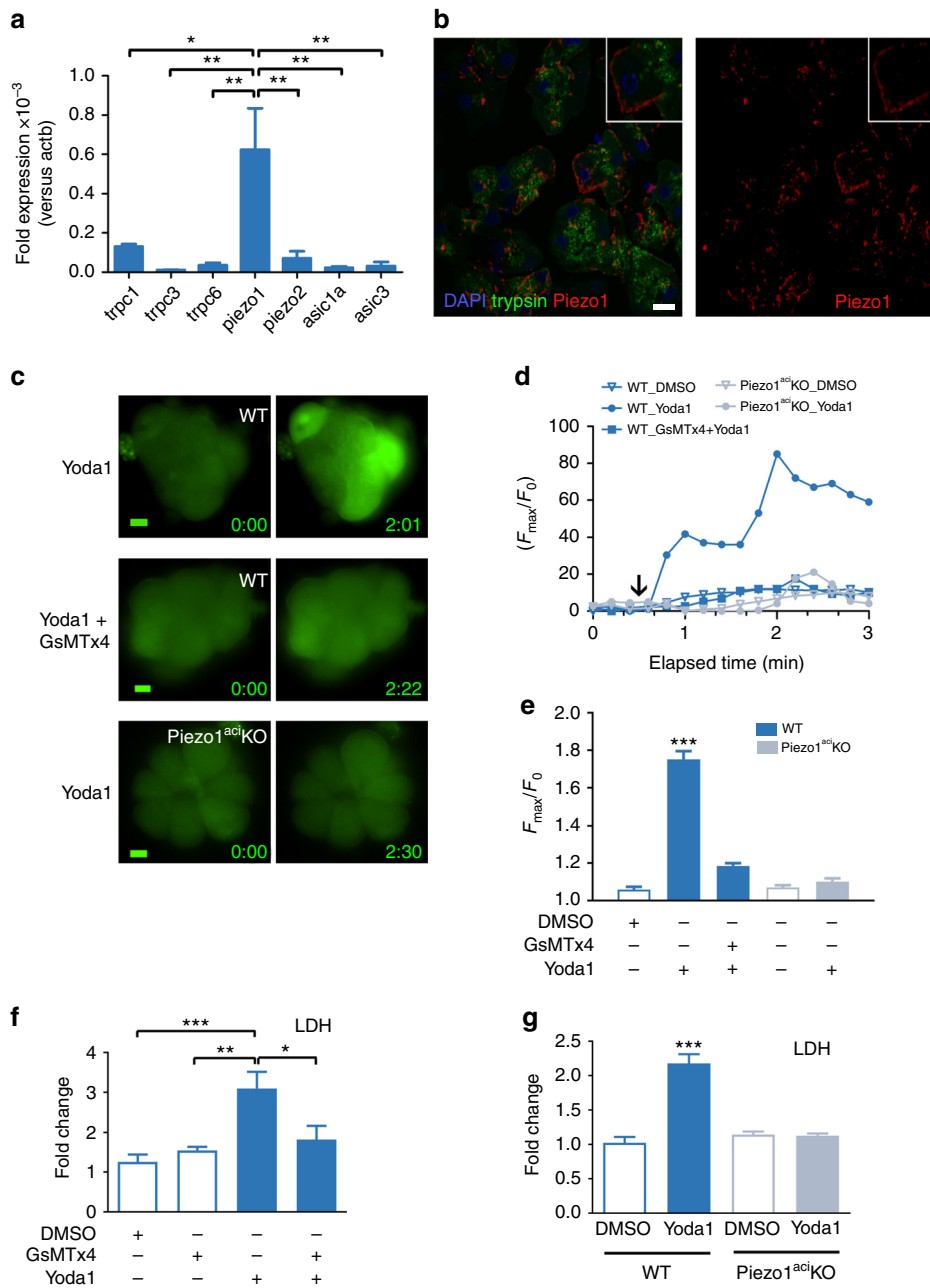

**Fig. 2** Piezo1 is expressed in pancreatic acinar cells. **a** The relative expression of mechanically activated ion channel messenger RNAs from C57BL/6J pancreatic acinar cells was analyzed by real-time RT-PCR using the $\Delta\Delta C_T$ method (biological $n = 3$). **b** Frozen sections (left panel) from mouse pancreatic tissue (C57BL/6J) were immunostained with antibodies against trypsin 3 (green) or Piezo1 (red). DAPI-stained nuclei are shown in blue. Inset shows a higher magnification of an acinar cell. Images were taken with a ×63 objective. Bar = 10 μm. **c** Representative live-cell imaging of pancreatic acini loaded with Calcium 6-QF at time 0 (left panel) and at the time of maximum fluorescence (right panel). Images obtained with a ×20 objective. Bar = 10 μm. **d** A representative time course of calcium fluorescence in acinar cells from littermate wild-type or Piezo1[aci] KO mice. Black arrow indicates when Yoda1 (25 μM) was added to the cells. **e** Statistical analysis of $F_{max}/F_0$ in acinar cells from littermate wild-type or Piezo1[aci] KO mice. GsMTx4 was added 2 min prior to Yoda1 ($n = 3$–5 experiments; 30–40 cells were analyzed for each experiment). LDH released from acini treated with Yoda1 (100 μM) from **f** C57BL/6J mice or **g** Piezo1[aci] KO and littermate (WT) mice ($n = 4$–5). Results are expressed as fold changes compared to the control condition (absence of test agents). DMSO was the vehicle for Yoda1; fold changes were determined vs. LDH release from cells that were incubated in absence of vehicle or compound. Statistical analysis was performed using ANOVA test with Tukey's post-test. *$P \leq 0.05$; **$P \leq 0.01$; ***$P \leq 0.001$

littermates lacking the *Cre* gene (WT), pancreatic acinar cells from Piezo1[aci] KO mice were insensitive to Yoda1 stimulation consistent with deletion of Piezo1 in these cells (Fig. 2d, e and Supplementary Movie 3). However, acinar cells of Piezo1[aci] KO mice exhibited a normal $[Ca^{2+}]_i$ response to cholecystokinin (CCK) stimulation (Supplementary Fig. 3) as has been previously observed in acini from WT animals[30].

**Yoda1 induces cell death in pancreatic acini through Piezo1 activation**. To confirm that Yoda1 was exerting its effects directly on Piezo1 in pancreatic acinar cells, we tested whether Yoda1 was capable of inducing cellular injury in isolated pancreatic acini in vitro by measuring lactate dehydrogenase (LDH) release. We first noted that Yoda1 in a dose-dependent manner increased LDH release in supernates of isolated acini from C57BL/6J mice

after 30 min of incubation at 37 °C (Supplementary Fig. 4). The effects of Yoda1 were blocked by pre-incubating acini in EGTA (2 mM)-containing buffer or pre-treating acini with GsMTx4 (2.5 μM) indicating that extracellular calcium is required for Yoda1's actions on acinar cells. GsMTx4 still protected acinar cells from cell death after 4 h incubation (Fig. 2f). To further explore the potential detrimental role of Piezo1 activation, we examined the effects of Yoda1 on acinar cells with genetic deletion of Piezo1. As shown in Fig. 2g, acini prepared from Piezo1[aci] KO mice lacking the functional Piezo1 channel were protected from Yoda1-induced damage.

**GsMTx4 reduces the severity of acute pancreatitis in vivo.** Having demonstrated that GsMTx4 would block $Ca^{2+}$ uptake in acinar cells, we then investigated whether GsMTx4 would alter the effects of pressure in an in vivo model of pressure-activated pancreatitis. Thus, GsMTx4 was administered intraperitoneally (270 μg/kg) 1 h prior to application of pressure into the pancreas. As shown in Fig. 3, all pancreatitis parameters induced by high intraductal pressure were improved by GsMTx4 pre-treatment indicating that the deleterious effects of pressure on the pancreas involved mechanosensitive ion channel activity.

**Genetic deletion of Piezo1 in acinar cells protects against pancreatitis.** In order to demonstrate the essential role of Piezo1 in developing acute pancreatitis, we examined the effects of high intraductal pressure in Piezo1[aci] KO mice. As shown in Fig. 4a and Supplementary Fig. 5, Piezo1 immunostaining was noticeably reduced on the surface of acini of Piezo1[aci] KO mice indicating that Piezo1 was deleted in most acinar cells. These animals were then used to determine the role of Piezo1 in mediating pressure-induced pancreatitis by subjecting tamoxifen-treated *Ptf1a-Cre[ER]; Piezo1[fl/fl]* (Piezo1[aci] KO) and *Piezo1[fl/fl]* (WT) mice to

intrapancreatic duct pressure. Pancreatitis was markedly reduced in Piezo1[aci] KO mice as indicated by reductions in edema, MPO measurements, and histological parameters. Histological scoring is a composite score for edema, neutrophil accumulation, necrosis, and hemorrhage. GsMTx4 inhibition in C57BL/6J mice was similar to the inhibition observed in Piezo1[aci] KO mice (see Supplementary Fig. 6).

**The Piezo1 agonist Yoda1 induces acute pancreatitis.** We next sought to determine if activation of Piezo1 using a specific Piezo1 agonist[12,31] in the absence of pressure could reproduce the effects we observed with high pressure. For these studies, the Piezo1 agonist Yoda1 (0.4 mg/kg) was infused into the pancreatic duct at a rate of 5 μL/min for 10 min so that fluid shear stress was negligible and intraductal pressure did not exceed 11 mm Hg (i.e., low pressure). Strikingly, all the pancreatitis parameters were significantly increased in Yoda1-treated WT littermates compared to the WT littermates treated with vehicle (Fig. 5a–d) alone.

In contrast, Yoda1 did not cause pancreatitis in mice with genetic ablation of Piezo1 in acinar cells. None of the parameters of pancreatitis, including edema, amylase, MPO, and histological scoring was elevated in Piezo1[aci] KO mice treated with Yoda1 (Fig. 5a–d). These studies demonstrate that pressure applied to the pancreas induces pancreatitis through Piezo1.

**Discussion**
Although it has long been recognized that the pancreas is sensitive to pressure and touch, a mechanism by which the pancreas can perceive such signals was unknown. The recent discovery that pressure sensitivity could be conveyed through cell surface, mechanically activated ion channels provided a novel mechanism by which mechanical force may mediate a variety of biological processes, including sensing touch and pain (somatosensation),

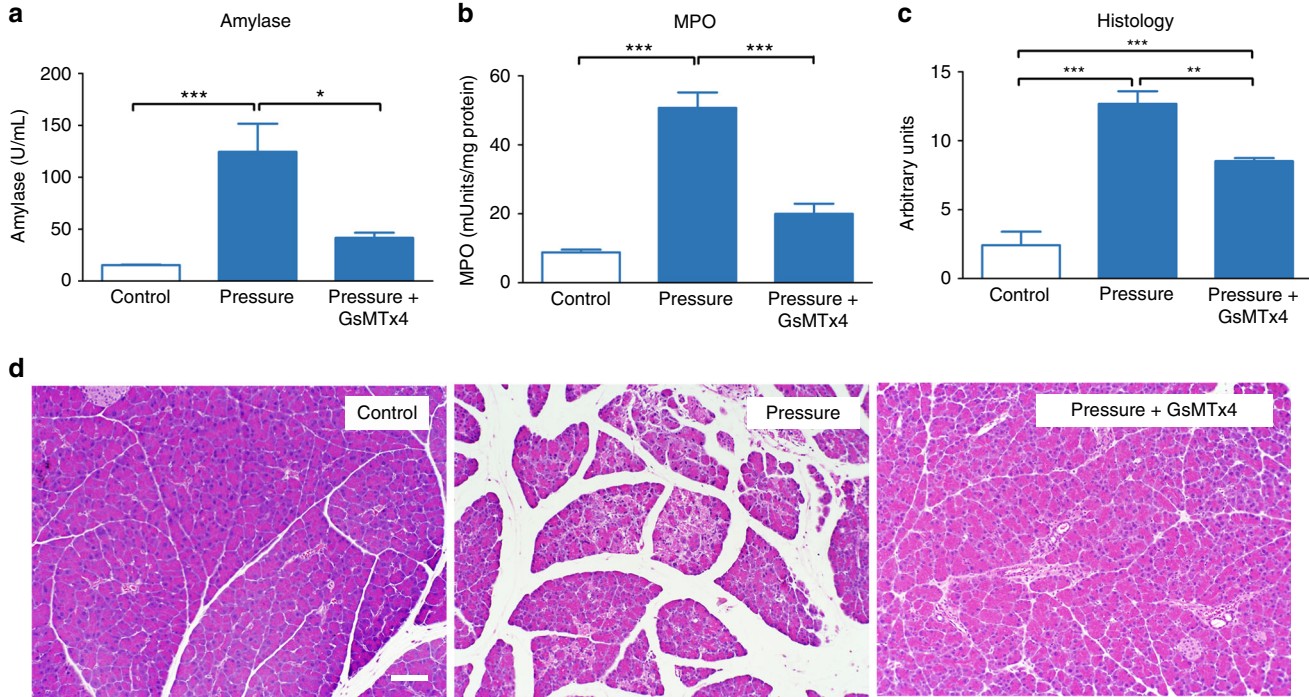

**Fig. 3** The Piezo1 antagonist GsMTx4 reduces the severity of pancreatitis. C57BL/6J mice were either sham-operated (control) or received intrapancreatic duct infusion of buffered saline at 25–33 mm Hg (pressure). GsMTx4 (270 μg/kg) was administered by intraperitoneal injection 1 h prior to intrapancreatic duct infusion of saline at 25–33 mm Hg (pressure + GsMTx4). **a** Blood amylase, **b** pancreatic MPO, and **c** histological scoring of pancreatitis severity were measured. Statistical analysis was performed using ANOVA test with Tukey's post-test. *P ≤ 0.05; **P ≤ 0.01; ***P ≤ 0.001 (n = 6). **d** Representative images of pancreatic tissue stained with H&E. Bar = 100 μm

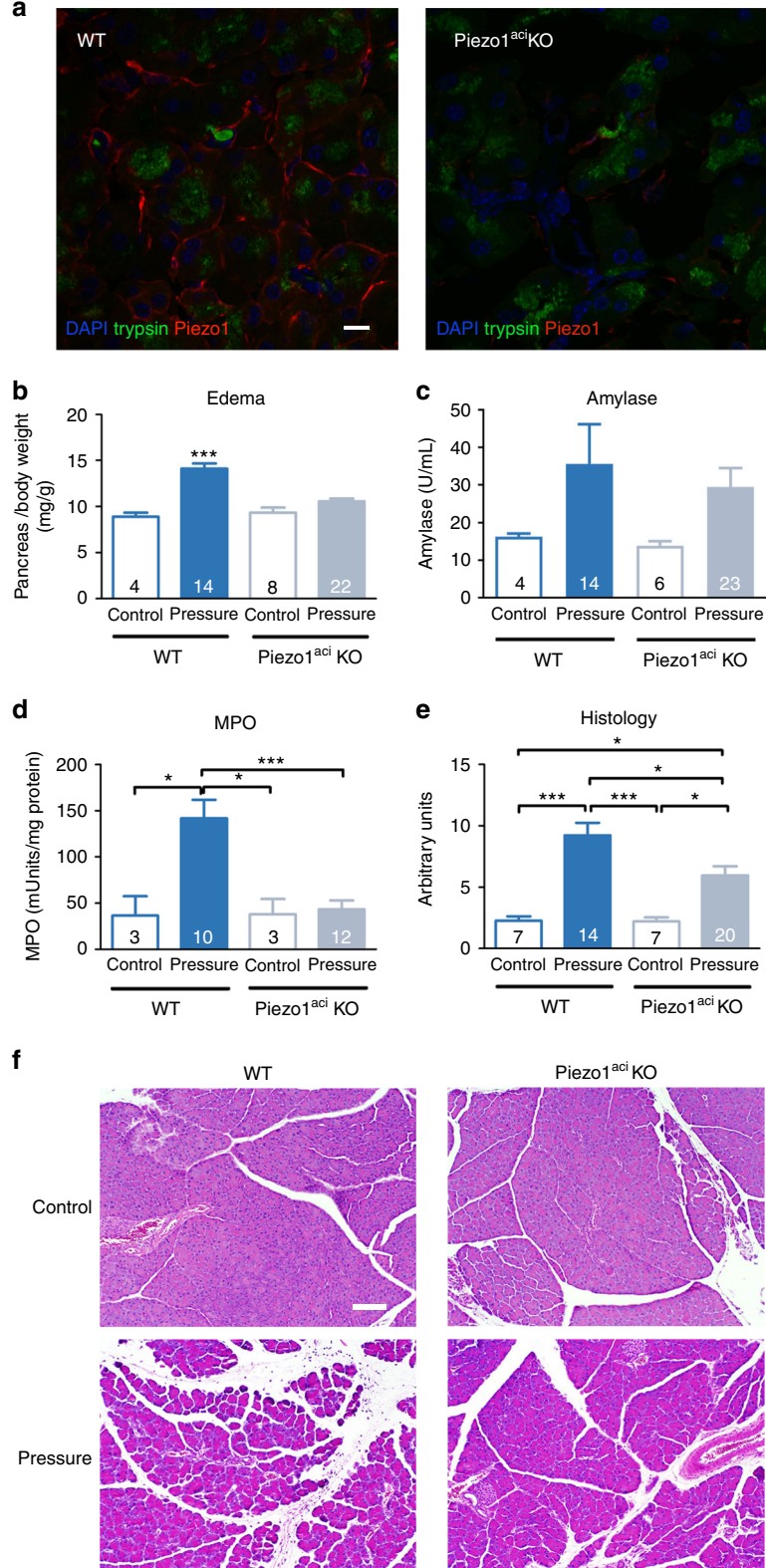

**Fig. 4** Genetic deletion of Piezo1 in pancreatic acinar cells reduces the severity of pancreatitis. **a** Confocal microscopic images of pancreatic sections from littermate wild-type (left panel) and Piezo1[aci] KO (right panel) mice imunostained for Piezo1 (red) and trypsin (green). DAPI-stained nuclei are blue. ×63 objective. Bar = 10 μm. Buffered saline was injected into the pancreatic duct at 25–33 mm Hg (pressure) of Piezo[aci] KO or littermate wild-type (WT) mice. Control animals were subjected to sham operation. Tissues were harvested 24 h post injection and analyzed for **b** pancreatic edema, **c** blood amylase, **d** MPO, and **e** histological scoring. The number of animals in each group is indicated in the respective column. Statistical analysis was performed using ANOVA test with Tukey's post-test. *$P \leq 0.05$; ***$P \leq 0.001$. **f** Representative images of pancreatic tissue stained with H&E taken with a ×10 objective. Bar = 100 μm

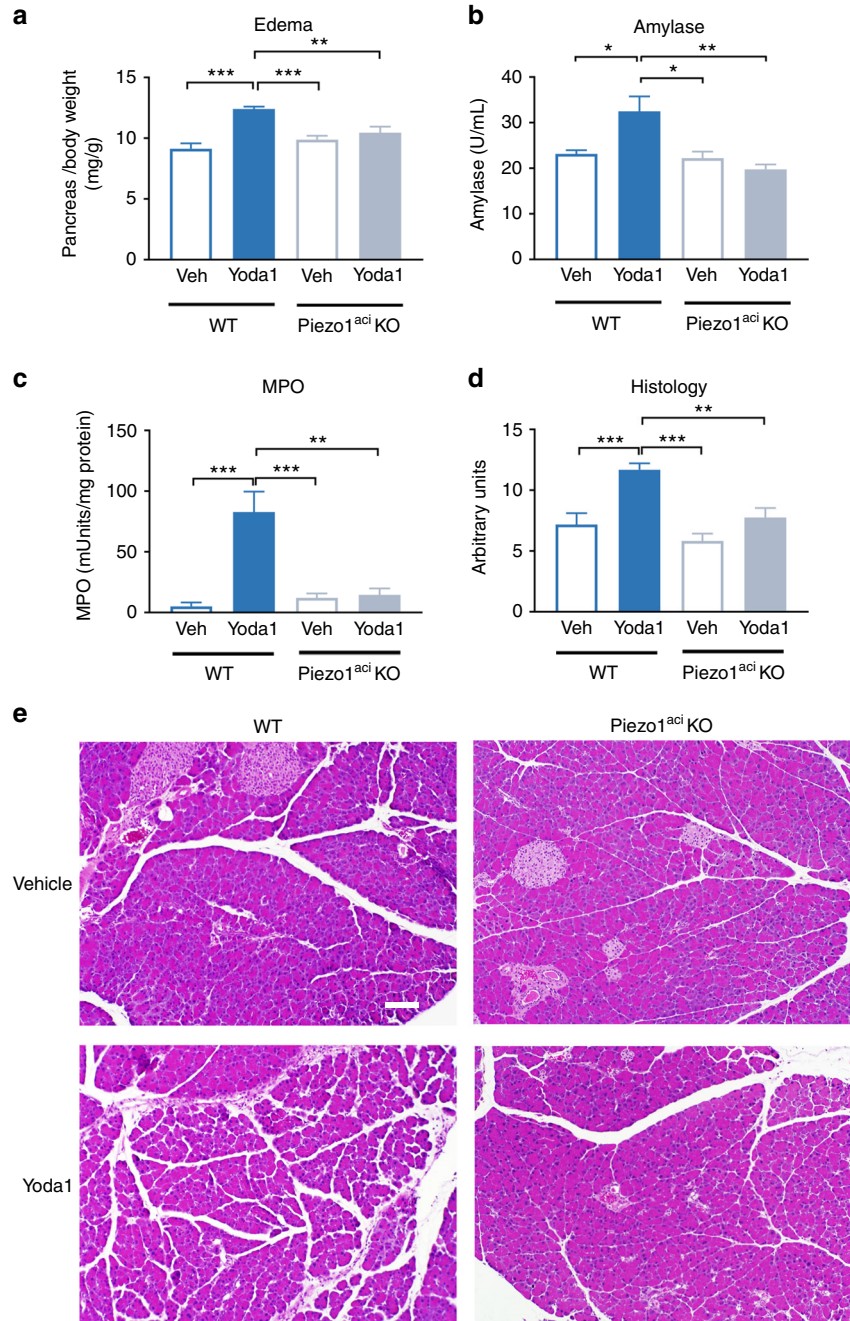

**Fig. 5** The Piezo1 agonist Yoda1 induces acute pancreatitis in wild-type mice but not in mice with genetic deletion of Piezo1. Yoda1 at a dose of 0.4 mg/kg in 50 μL (1.1 % DMSO; 4.8% ethanol; 94.1% buffered saline, pH 7.6) was infused into the pancreatic duct of WT or Piezo1[aci] KO mice over 10 min at pressures not exceeding 7–11 mm Hg. Either vehicle or Yoda1 was infused. Tissues were harvested 24 h post injection and analyzed for **a** pancreatic edema, **b** serum amylase, **c** pancreatic myeloperoxidase (MPO), and **d** histological scoring. $n = 6$–8 animals per group. Statistical analysis was performed by ANOVA with Tukey's post-test. $*P \leq 0.05$; $**P \leq 0.01$; $***P \leq 0.001$. **e** Representative images of pancreatic tissue stained with H&E are shown. Bar = 100 μm

sound (hearing), and shear stress (blood pressure)[32,15]. This discovery coupled with the observations that mechanical forces cause pancreatitis led us to ask if mechano-transducing ion channels exist in the pancreas. First, we examined the gene expression of known putative mechanically activated ion channels in mouse pancreas and discovered that *Piezo1* was the predominant mechanically activated ion channel expressed in the pancreas. Piezo1 immunostaining was seen in pancreatic acinar cells and was concentrated along the surface of cells. These findings laid the groundwork for evaluating the possible role of Piezo1 in mechano-transduction in pancreatic acinar cells.

As an exocrine organ the pancreas has limited ways it can respond to physiological or pathological stimuli. Over 90% of the pancreas is comprised of acinar cells that produce abundant digestive enzymes that are synthesized as inactive precursors or stored in zymogen granules. Injury to the pancreas disrupts normal intracellular calcium signaling leading to abnormal zymogen and lysosomal granule fusion and premature activation of enzymes. Activation of zymogens within the gland initiates autodigestion and is a key feature of pancreatitis pathogenesis.

Piezo proteins are pore-forming proteins that reside in the plasma membrane and possess cation channel properties[7].

Mechanical stimulation causes the pore to open allowing cations such as $Ca^{2+}$ to flow into the cell, along the ion's concentration gradient[33]. In the pancreas, intracellular calcium is a key regulator of exocytosis and is tightly controlled by a balance of release from intracellular stores, extrusion via calcium pumps, and influx through store-operated entry pathways[34,35]. Pathological stimulation to the pancreas that causes pancreatitis, evokes sustained global elevations of cytosolic $Ca^{2+}$, inhibition of pancreatic enzyme secretion, intracellular protease activation, and is linked to cell fate. Perturbations of $Ca^{2+}$ signaling have been associated with both apoptotic and necrotic cell death[36]. Thus, excessive cytoplasmic calcium signals are toxic to pancreatic acinar cells and are responsible for initiating the intracellular events leading to protease activation and autodigestion resulting in cellular necrosis and pancreatitis. Piezo1-facilitated calcium entry is not dependent upon an initial intracellular signal and high Piezo1 activation causes sustained $[Ca^{2+}]_i$[12] that could be detrimental in pancreatic acinar cells.

Our observations with Yoda1 support this notion since activation of Piezo1 with Yoda1 produced pancreatitis in vivo. Blockade of Yoda1's effects on acinar cells with GsMTx4 suggests that Piezo1 activation is the inciting event in the pancreatitis cascade. These findings are further substantiated by the demonstrations that both pharmacological inhibition and genetic deletion of Piezo1 inhibited pressure-induced pancreatitis. The slight variability in some pancreatitis parameters we observed in in vivo experiments could be attributed to the genetic backgrounds of the mice. However, the experimental design was also slightly different since unlike the C57BL/6J mice, both Piezo1[aci] KO and their WT littermates were subjected to tamoxifen treatment 8 days before Yoda1 or pressure-induced surgery. C57BL/6J mice were used in experiments described in Figs. 1 and 3, while Piezo1[aci] KO and their littermates were used for experiments presented in Figs. 4 and 5. Nevertheless, the finding that pharmacological or genetic deletion of Piezo1 in mice of slightly different backgrounds exhibited similar patterns of protection against pancreatitis reinforces the concept that Piezo1 mediates pressure-induced pancreatitis.

Gallstones are a common cause of acute pancreatitis and it is believed that this occurs through blockade of the ampulla of Vater by migrating gallstones that increase pancreatic duct pressure[37]. In those instances when the gallstone persists in the duct, endoscopic removal could attenuate the severity of pancreatitis and improve clinical outcomes by relieving pancreatic duct occlusion[38,39].

Elevated pancreatic duct pressure has been demonstrated in patients with chronic pancreatitis[40], which may be due to increased viscosity of pancreatic juice[41,42]. However, it was unclear whether chronic pancreatitis arises from or leads to increased pancreatic duct pressure until it was demonstrated experimentally that sustained elevation of pancreatic duct pressure in rats caused chronic pancreatitis[43,44]. These studies support the paradigm by which the pancreas can sense pressure and elevated pressure can cause pancreatitis. Our observation that Piezo1 activation directly damages acinar cells raises the possibility that elevated pressure within the pancreas contributes to the progression of pancreatitis through a Piezo1-mediated mechanism.

As chronic pancreatitis progresses, alterations in pancreatic duct anatomy develop. These range from tiny obstructions of small branches of the pancreatic duct caused by precipitation of proteinaceous fluid as seen in hereditary, idiopathic or alcoholic pancreatitis[45,46] or gross changes in larger ducts of the pancreas produced by duct strictures, intrapancreatic duct stones, or pseudocysts compressing on the pancreatic duct[47]. In these cases it is likely that the increased intrapancreatic duct pressure not only disturbs normal pancreatic acinar function but directly damages acinar cells thus promoting pancreatitis.

Acute pressure causing injury to the pancreas can be associated with several clinical conditions. Trauma to the pancreas occurs in 3–5% of patients with abdominal injury[48]. Pancreatitis also occurs with simple manipulation of the gland at the time of surgery when no direct injury is involved. Injection of fluid into the pancreatic duct with ERCP produces pancreatitis in 6–10% of high-risk individuals[49]. An underlying trait common to these diverse modes of pancreatitis is the application of pressure to the gland—either indirect (abdominal trauma) or direct pressure to the pancreas (surgical manipulation) or pressure within the pancreas (ERCP). Recognizing that the pancreas possesses pressure-sensitive, mechanically activated ion channels now provides a mechanism by which these injuries occur. Moreover, our current findings suggest that strategies to block Piezo1 could be used to prevent pancreatitis when manipulation of the pancreas is anticipated as with pancreatic surgery or ERCP.

## Methods

**Animals.** WT animals: C57BL/6J male mice 6–8 weeks old (Jackson Labs) were used for in vivo experiments and for the isolation of pancreatic acini.

Piezo1 knockout mice: The $Ptf1a^{tm2(cre/ESR1)Cvw}/J$ mice (Jackson Labs) were crossed with $Piezo1^{fl/fl}$ mice (generous gift from A. Patapoutian[31] to generate the line $Ptf1a^{CreERTM}$; $Piezo1^{fl/fl}$). $Piezo1^{fl/fl}$ (referred to as WT) and $Ptf1a^{CreERTM}$; $Piezo1^{fl/fl}$ mice when 4 weeks of age were subjected to five daily intraperitoneal injections of 1 mg tamoxifen. The $Ptf1a^{tm2(cre/ESR1)}$; $Piezo1^{fl/fl}$ mice after Cre induction express a truncated Piezo1 exclusively in pancreatic acinar cells. We refer to tamoxifen-treated $Ptf1a^{CreERTM}$; $Piezo1^{fl/fl}$ mice as Piezo1[aci] KO. Littermates were used for experiments and surgery was performed 8 days after the last tamoxifen injection. Genomic DNAs from pancreas harvested at the end the experiments were analyzed by PCR. A band indicative of the genetic ablation by Cre was detected in the Piezo1[aci] KO DNAs[31]. $Ptf1a^{CreERTM}$; $Piezo1^{fl/fl}$ animals lacking the KO band were excluded from analysis. Both male and female mice aged 7–10 weeks were used in experiments involving genetic modification of Piezo1.

EYFP expression in acinar cells of Piezo1[aci] KO mice: The $Ptf1a^{tm2(cre/ESR1)}$; $Piezo1^{fl/fl}$ mice were crossed with $Ptf1a$-$Cre^{ER}$; $Gt(ROSA)26Sor^{tm3(CAG-EYFP)}$; $Piezo1^{fl/fl}$ to obtain the EYFP;Piezo1[aci] KO mouse line. These mice after tamoxifen induction express EYFP and truncated Piezo1 in acinar cells.

Mice were housed in a 12:12-h light-dark cycle and given water and chow ad libitum. Studies were approved by the Institutional Animal Care and Use Committee of Duke University.

**Materials.** GsMTx4 was purchased from Abcam; CCK8 was purchased from Sigma-Aldrich. Yoda1 was purchased from R&D. The LDH cytotoxicity kit was purchased from Promega.

**Surgical procedures.** Surgery was performed as previously described[50] with the following modifications. Pressure experiments were conducted by injecting buffered (HEPES 10 mM, pH 7.6) isotonic saline solution into the pancreatic duct using a high performance PHD ultra-syringe I/W programmable pump with a pressure transducer APT300 Hg from Harvard Apparatus that was set to a range of 0–200 mm Hg. Intraductal pressure was monitored continuously during injection using the software dedicated to the instrument. The low-pressure condition was produced by infusing 50 µL of buffered saline at a rate of 5 µL/min for 10 min. The high-pressure condition was produced by infusing 400 µL of buffered saline at 80 µL/min for 5 min.

Yoda1 at a dose of 0.4 mg/kg in 50 µL (1.1% dimethylsulfoxide (DMSO); 4.8% ethanol; 94.1% buffered saline), was injected using the same procedure described above at a rate of 5 µL/min for 10 min. Yoda1 was first dissolved in DMSO at a concentration of 50 mM. It was then diluted to 10 mM in ethanol. This solution was then diluted in buffered saline (pH 7.6) at 37 °C following the ratio indicated above. Yoda1 remained in solution in this condition.

Methylene blue (Fluka cat. 03978) was added to the solution during the infusion to monitor the surgical procedure. Animals were excluded from the study if leakage or intestinal ischemia were observed.

Control animals also underwent laparotomy and the pancreas and pancreatic duct were identified but no pressure was applied. The abdomen was closed in two layers.

At the conclusion of the in vivo experiments, the entire pancreas was removed and weighed. The mid-region of the pancreas corresponding to the gastric and duodenal lobes and weighing 100–125 mg was carefully isolated and used for histological and biochemical assessments. A section of this region was taken for histological processing. The entire histological section was used for scoring.

**In vitro cell studies**. Isolated pancreatic acini and acinar cells were prepared as previously described[51,52]. Acini for LDH were prepared by using a modified KHB buffer (glucose 200 mg, 2 mL of essential amino acids 50× (Gibco catalog number 11130-051), glutamax 100× (Gibco catalog number 35050-061), 6.5 mL of 0.5 M NaHCO$_3$, 4.65 mL of 2.38 M NaCl, 0.5 mL of 0.94 M KCl, 0.5 mL of 0.2 M Na$_2$HPO$_4$, 0.5 mL of 0.23 M MgCl$_2$, 0.2 mL of 0.51 M CaCl$_2$, and 10 mg of soybean trypsin inhibitor (SBTI 1-S) (Sigma, catalog number T9003) for a final volume of 100 mL. Solutions A, B, and C were prepared by addition of bovine serum albumin (BSA) 20 mg (A), 100 mg (B), and 1 g (C) to a respective volume of modified KHB buffer (A and B: 10 mL; C: 25 mL). A unit of 2 mg of collagenase NB 8 (SERVA, catalog number 17456) was dissolved solution A. A volume of 5 mL of this solution was used to inflate the pancreas, which was incubated for 10 min in a shaking bath at 37 °C. Solution A was then replaced by fresh solution A with collagenase (5 mL). The tissue was incubated for 45 min in a shaking water bath at 37 °C. The tissue was then disrupted by repeated pipetting using first a pipet with a 2 mm opening (10×) and then a pipet with 1.5 mm opening (5×). The tissue was then filtered through a 250 μm nylon filter. The filter was washed with 10 mL of solution B. The filtrate was then loaded onto solution C (filtrate 4 mL/solution C 8 mL). The cells were centrifuged in a IEC clinical centrifuge at 200 rpm for 5 min. The cell pellet was washed with DMEM/F12 media (Gibco, catalog number 11330-032) with 0.3% BSA and SBTI (0.01%). The final cell pellet was resuspended in 30 mL of the same media with BSA and SBTI inhibitor. Cells were incubated for 30 min at 37 °C with very low shaking. Cells were aliquoted onto 24-well plates. Reagents were then added and the cells were incubated in a tissue culture CO$_2$ incubator. Acini responsiveness to supramaximal doses of CCK was used to monitor our preparations[53]. Yoda1 was resuspended in 50 mM DMSO and aliquoted in warm media to the final concentration.

For live-cell imaging studies, acini were prepared with a slight modification: the tissue was incubated for 30 min instead 45 min; the tissue was filtered through a 100 μm nylon filter and the cell pellet was washed with Leibovitz's media (Gibco catalog number 1145064).

Acinar cells were prepared following the established protocol[54] with slight modifications: collagenase NB 8 (3 mg/15 mL in Gey's balanced salt solution) was used in the absence of pronase and DNase I. At the end of the preparation, the pellet containing the acinar cells was washed with PBS and centrifuged at $450 \times g$ twice for 10 min. The cells were then used for RNA isolation.

**Assays**. Blood amylase: Blood collected from decapitated mice was centrifuged at $800 \times g$ for 10 min at room temperature. The serum amylase concentration was measured by colorimetric method using the Phadebas amylase test tablet (Magle Life Sciences, Cambridge, MA). A standard curve was prepared using α-amylase (Sigma, catalog number A3176). Extracts and standards were incubated at 37 °C for 15 min in 1 mL of sodium phosphate buffer containing 20 mM NaH$_2$PO$_4$, 50 mM NaCl, and 0.05% NaN$_3$ at pH 7.4. The reaction was stopped by adding 4.3 mL of NaOH at 35 mM. The sample mixture was centrifuged for 15 min at 4 °C at 1900 × $g$[50]. The optical density (OD) of the supernate was measured at 620 nm. Serum amylase levels were expressed in mU/mL.

MPO assay: MPO was measured using the methods previously described[55,56]. Pancreatic tissue (100 mg) was homogenized in 1 mL of 100 mM phosphate buffer (pH 7.4) containing protease inhibitor cocktail (Roche, complete and EDTA-free; catalog number 11873580001). The tissue samples were homogenized with a Brinkmann polytron, then centrifuged at 16 000 × $g$ for 15 min at 4 °C. The pellets were washed twice with phosphate buffer (pH 7.4). The pellets were resuspended in 100 mM phosphate buffer (pH 5.4) containing 0.5% hexadecyltrimethyl ammonium bromide (Sigma catalog number H5882) 10 mM EDTA, and the protease inhibitor cocktail. The pellet was further homogenized and subjected to three cycles of sonification, freezing, and thawing. The extract was then centrifuged as before and the supernate used for MPO assay. MPO levels were measured by a colorimetry method using the substrate 3,3′,5,5′ trimethylbenzidine (TMB) at 40 μM final concentration with H$_2$O$_2$ (0.001% final concentration) as cofactor. Supernates and MPO standards (Sigma, M6908) were preincubated in the presence of TMB for 3 min at 37 °C. Hydrogen peroxide was then added and incubated for 3 min at 37 °C. The enzymatic reaction was stopped with acetate buffer 0.2 M (pH 3.2). ODs were read at 655 nm. Concomitantly, protein concentration of the supernates were measured using the Micro BCA Protein Assay Kit (Thermo Scientific Kit, catalog number 23235). MPO levels were expressed in mU/mg protein. Hematoxylin-and-eosin-stained tissues were imaged on an Olympus Vanox AHBS3 photomicroscope, with an Olympus DP70 digital camera, using a PlanApo ×10 objective. RNA isolation from acinar cells, reverse transcriptase, and real-time PCR procedures were also previously described[50]. RNAs were isolated using the RiboPure Kit (Invitrogen, catalog number AM1924) followed by DNase1 digestion (Invitrogen, catalog number AM1906). RNAs (2 μg) were reverse transcribed into cDNA using the High Capacity cDNA reverse Transcription Kit (Applied Biosystems; catalog number 4368814). Validated TaqMan probes were purchased from ThermoFisher.

**Immunostaining and microscopy**. Frozen sections of pancreatic tissue were incubated with a rabbit anti-Piezo (Alomone; #APC-087; 1:300) and/or a mouse anti-trypsin 3 (R&D; AF3565; 1:400) antibody. The signal from Piezo1 immunostaining was amplified by tyramide signal amplification following the manufacturer's recommendations (Life Technologies; #T20924). All images were taken with a Zeiss 780 confocal microscope with either a ×20 objective or a ×63 oil-immersion objective.

**Calcium imaging**. Pancreatic acini were plated on glass coverslips pre-treated with Cell-Tak (Corning). Cells were loaded Calcium 6-QF (Molecular Devices) in minimum essential medium:Hanks' balanced salt solution (MEM:HBSS; 1:1) for 1 h at 37 °C in a CO$_2$ incubator. Prior to imaging, MEM:HBSS media were replaced by HBSS with 2 mM Ca$^{2+}$. Imaging was performed at room temperature using a Zeiss Axio observer Z1 with a ×20 objective. Images were recorded every 400 ms over 3 min and data were analyzed using MetaMorph software (Molecular Devices). Yoda1 resuspended in DMSO at 50 mM was added at a final concentration of 25 μM in HBSS media. Yoda1 was added 30 s from the start of the recording. When GsMTx4 was tested on acini, the cells were preincubated with GsMTx4 (2.5 μM) 2 min prior to recording. Overloaded or faintly fluorescent acini ($F_0 < 150$ AI) were excluded from the analysis as well as cells demonstrating unstable fluorescence before loading.

**Statistical analysis**. Results were expressed as mean ± SEM. Mean differences between two groups were analyzed by Student's $t$-test and mean differences between multiple groups were analyzed by one-way analysis of variance with the Tukey's multiple comparison post-test (GraphPad; prism 5.03). $P$ values < 0.05 were considered significant. *$P \leq 0.05$; **$P \leq 0.01$; ***$P \leq 0.001$.

**Data availability**. The data that support the findings of this study are available from the corresponding author upon reasonable request.

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

## Acknowledgements

This work was supported by NIH grants R01 DK064213, DK098796, and the Department of Veterans Affairs BX002230. We thank Dr. Venkata Jabba for technical advice. We acknowledge generous assistance from the Duke University Light Microscopy Core and Transgenic Core Facilities.

## Author contributions

J.M.-J.R. designed and performed experiments, analyzed the data, and wrote the manuscript. R.A.S. designed and performed experiments, analyzed the data, and reviewed the manuscript. S.M.S. assisted in the design, performed experiments, analyzed the data, and reviewed the manuscript. S.R.V. provided critical review of the data. R.A.L. designed and supervised the project, wrote the manuscript, and provided funding for the project. All the authors reviewed the manuscript

## Additional information

**Competing interests:** The authors declare no competing interests.

