## [Peer Review File · Nature Communications]

Reviewers' comments:

Reviewer #2 (Remarks to the Author):

The manuscript by Romac, Shahid et al., (NCOMMS-17-19425-T) describes experiments aimed at determining whether Piezos or putative mechanoactivated channels underlie the induction of pancreatitis by pressure applied to the pancreatic duct. The authors make a good case that Piezo1 is functionally expressed on pancreatic acinar cells. They present, for the first time, an in vivo model in which pressure via the pancreatic duct causes acute pancreatitis by measuring critical parameters known to be observed in this disease (serum amylase, pancreatic MPO, and histological evidence (but see below #1 for concerns)). They go on to test whether Piezo1 mediates the observed edema, MPO and histological changes (but amylase not tested!) and show data that suggests this is the case (but see below). Pressure-induced acute pancreatitis (as quantified by serum amylase, the presence of MPO myeloperoxidase which is indicative of neutrophil infiltration, histological features and edema in some cases) was suppressed by the MA channel inhibitor GSMTx4 (Fig3) and required Piezo1 (at least in part). Acute pancreatitis was shown to be induced by the Piezo1 agonist Yoda1. Importantly, the effects of Yoda1 were partially dependent on Piezo1 expression in acinar cells, a critical finding that was surprisingly presented in supplemental information (Suppl. Fig 6), perhaps because of the variability observed (see below #3).

Although the authors claim Piezo1 activates inflammatory pathways (Fig5) and cell death (Suppl Fig 4) by using (high) concentrations of Yoda1 applied to acinar cells in vitro, it is not clear that these processes could be initiated by activation of endogenous Piezo1 in an in vivo situation since they did not test Yoda1 effect's on KO animals (GsMTx4 is not specific for Piezo1).

Nonetheless, this work sheds light on the molecular mechanisms that underlie pressure induced pancreatitis and reveals yet another organ system in which Piezos appear to play a role in pathophysiology.

Major concerns.

1. The in vivo pressure-induced pancreatitis model requires additional information for validation. It is not apparent that the low mechano (5ul/min x 10 min= 50ul total infusion) is an adequate control for the high mechano (80uL/min x 5 min= 400uL total infusion) stimulation that produces pancreatitis-like symptoms.

a. The extent of infiltration of the dye in the high mechano condition ("pressure") is not complete but it is hard to tell the extent from figure 1a. The reproducibility and extent of infiltration needs to be determined, especially since there is significant variability (see #3) that calls into question the conclusions. Is Fig1a taken immediately after the end of the injection or is it 24 hours after injection?

b. What is the extent of infusion into the pancreas during 5ul/min x 10min ("control")? If the low condition only affecting the first 20-30% of the pancreas during the injection time, this calls into question how this could be a true "control" for the larger injection. It also begs the question: where are the sections taken for histological analysis? This should be clearly stated and sections from different regions (proximal/distal to injection site; gastric, splenic, duodenal lobes) should be compared. The control sections need to be taken at the same location as experimental sections in comparison to the injection site. "Portions of the pancreata" is not specific enough. Where exactly were the sections taken? Are the sections consistent throughout the pancreas?

2. Piezo1 underlies the acute pancreatitis observed by pressure in vivo. The critical figure 6 is problematic.

a. variability of control and pressure induced levels between figures 1,3 and 6 are cause for serious concern. Although the variability in basal conditions/fold increase does not appear to be due to strain background, it is not clear and should be considered and discussed. For instance, MPO control values in WT are ~4-fold higher n fig 6c compared to figs 1 and 3. The pressure

induced MPO is at least 3-fold higher than that in figs 1 and 3.

b. The exact genotypes of the mice used in these studies is not clear. On p12 the fl/fl mice were defined as wildtype but then on p 13 "littermates were used for experiments". Are the control "wildtype" mice really Cre-;Piezo1fl/fl mice? It should be clearly stated what genotype/background of mice were used in each experiment.

c. what sections are compared for control and pressure?

d. why was serum amylase not measured in these experiments?

e. what do the n refer to in the bars in fig 6d? Does n= the number of sections from a single animal or number of animals from which a certain number of sections were scored? Or a combination? The number of separate animals and the number of sections from each animal needs to be clearly stated.

3. Histopathology scoring. Since the GSMTx4 (fig3) and Piezo1aciKO (fig6) "reversal" of pressure induced damage is only partial, it is imperative to give values for each of the parameters scored. Is GsMTx4 or cKO partially reversing each of the parameters similarly or is GsMTx4 or cKO fully reversing some of the parameters (Piezo1 dependent) and having no effect on other parameters (Piezo1 Independent)? For instance, is edema and infiltration always dependent on Piezo1 but fat necrosis not Piezo1 dependent?

Histological parameters should be explicitly described and any differences from the 1989 paper should be discussed. The reference given refers to an earlier work by these authors in which the parameters are still not explicitly spelled out and refers the reader to yet another earlier, 1989 paper.

4. In the abstract the authors claim: "to determine whether [the application of pressure within the gland] was through Piezo1 activation, we induced pancreatitis by intrapancreatic duct instillation of ... Yoda1." To this reviewer, the Yoda1 experiments are weak. It is very concerning that Yoda1 infusion under low mechano stimulus appears to induce very variable effects in the pancreatitis parameters. Comparing Fig 4 with Suppl Fig6 suggests the observed variability may negate the significant effects shown in Fig. 4. For instance, with amylase: 10x increase observed in Fig 4b but only 2-3 fold in WT in Suppl Fig 6b. The cKO animals reveal a 2-3 fold increase in the presence of Yoda1. The baseline differences are substantial. The in vitro experiments on acinar cells indicate that there is a late (at >~1 min) increase in intracellular calcium by 25uM Yoda1 that is only partially piezo1 dependent. Taken together, the data indicate issues with variability and off-target effects of Yoda1.

a. What is the extent of the Yoda1 infusion throughout the pancreas? This is not disclosed. Concentration of Yoda1 used is very high; at 100uM, Yoda1 crashes out of aqueous solution. Although the Yoda1 concentration used in vitro is 25 uM (fig 2), it is not clear what Yoda1 concentration is used in Fig5B,C.

b. The biggest concern is the use of a very high concentration of Yoda1 that likely would be activating more endogenous Piezo1 than would be activated by physiological pressure stimuli. Using the pressure model, are inflammatory mediators and mRNA changed in wildtype controls but not Piezo1cKO?

c. Histology is incompletely reversed in cKO in pressure model, but is completely reversed in cKO in YODA1 induced acute pancreatitis. On the other hand, the reverse seems to be true for edema and MPO in Fig 6 (pressure model). Please discuss.

d. Appropriate statistical analyses need to be provided for all relevant comparisons.

5. If the set of experiments used to conclude that Piezo1 activation induces inflammatory pathways is included in the manuscript it is imperative to show that the effects of Yoda1 on Ikb and all the genes shown in 5B and 5C are dependent on Piezo1 activity using the cKO animals vs

appropriate Cre- littermate controls. There appear to be issues with these experiments, for instance, Fig5a,b suggests reduction in I κ B but there is no housekeeping (actin?) control for normalization.

6. Statistical analyses are not included.

Fig5c: is NORMALIZED I κ B reduction by 10uM yoda1 significant?

Fig 6d: what is the n referring to? How many sections were scored from how many different animals?

Suppl Figure 4a: is 10uM yoda1 significantly increased compared to vehicle?

Suppl Figure 4b: is 100uM yoda1 effect in presence of EGTA or GsMTx4 significantly different from EGTA or GsMTx4 controls?

Suppl Figure 6b and 6c: in cKO animals, is Yoda1 (700uM) effect signif different from veh control?

Minor.

Fig1a. what is the concentration of methylene blue used?

Fig 1: why wasn't edema measured in these experiments?

Fig2e. why is DMSO applied when Yoda1 is added if this is the vehicle for Yoda1?

Fig6. Include plasma amylase results for pressure model using Piezo1 cKO animals.

Suppl fig 4a: Was the LDH release from isolated acini at 10uM Yoda1 significantly different from vehicle?

Methods p. 13: EYFP;Piezo1aciKO are stated to express truncated Piezo1 in acinar cells. Is this predicted or tested experimentally?

Reviewer #3 (Remarks to the Author):

This is a highly novel and important finding with significant clinical implications.

Liddle et al have followed up on the recent discovery of Piezo-1 and Piezo-2, which are pressure activated ion channels in the etiopathogenesis of acute pancreatitis.

They have demonstrated convincingly that Piezo-1 is expressed in pancreatic acinar cells and that pathophysiological Piezo1 activation is associated with pressure-induced acute pancreatitis, whilst blocking of Piezo1 channels and targeted Piezo-1 deletion abrogated the effects of pressure in the induction of acute pancreatitis.

They could have gone on to show that acute pancreatitis could still be induced by other trigger factors such as bile acids and/or fatty acid ethyl esters in Piezo-1 deleted/blocked acini.... additive? synergistic? etc.

Certainly this should be picked up in the discussion. In gallstone pancreatitis there is always the passage of multiple gallstones into the duodenum and this will inevitably result in both the temporary obstruction to both the main bile duct and main pancreatic duct and then bile reflux

into the main pancreatic either directly or from the duodenal lumen. Do bile acids promote acute pancreatitis under low pressure conditions (ie just above normal pressure) ?

Acosta JM, Ledesma CL. Gallstone migration as a cause of acute pancreatitis. N Engl J Med. 1974;290(9):484-487;

Neoptolemos JP et al. Results of a prospective randomized trial of ERCP and endoscopic sphincterotomy in acute gallstone pancreatitis. Lancet 1988; ii: 979-983;

van Santvoort HC et al. Early endoscopic retrograde cholangiopancreatography in predicted severe acute biliary pancreatitis: a prospective multicenter study. Ann Surg. 2009;250(1):68-75.

Thus the therapeutic importance of blocking Piezo-1 would be considerable in acute pancreatitis.

Moreover there needs to be some discussion about chronic pancreatitis – hereditary, idiopathic, and other (alcohol etc). This is a progressive disease promoted not only ongoing trigger factors (genetic mutations, smoking etc) but also by gross and micro degrees of obstruction.

This study would promote the importance of free drainage of major ducts but also of Piezo-1 blockade to lessen the impact of intermittent small duct obstruction.

Finally in the Introduction they mention that abnormally high intracellular calcium concentrations lead to impaired zymogen granule processing and fusion of zymogen granules with lysosomes causing trypsinogen activation.

But Raraty et al showed that trypsin activation and vacuolization are caused directly by the rise in intracellular calcium concentration without the requirement for co-localization.

Ref 17. Raraty, M. et al. Calcium-dependent enzyme activation and vacuole formation in the apical granular region of pancreatic acinar cells. Proc Natl Acad Sci U S A 97, 13126-13131 (2000).

Reviewer: John Neoptolemos

Reply to Reviewers

We would like to thank the reviewers for their thoughtful consideration of our manuscript and suggestions for its improvement. We have incorporated these recommendations into our revised manuscript as outlined below.

Reviewer #2 (Remarks to the Author):

The manuscript by Romac, Shahid et al., (NCOMMS-17-19425-T) describes experiments aimed at determining whether Piezos or putative mechanoactivated channels underlie the induction of pancreatitis by pressure applied to the pancreatic duct. The authors make a good case that Piezo1 is functionally expressed on pancreatic acinar cells. They present, for the first time, an in vivo model in which pressure via the pancreatic duct causes acute pancreatitis by measuring critical parameters known to be observed in this disease (serum amylase, pancreatic MPO, and histological evidence (but see below #1 for concerns)). They go on to test whether Piezo1 mediates the observed edema, MPO and histological changes (but amylase not tested!) and show data that suggests this is the case (but see below). Pressure-induced acute pancreatitis (as quantified by serum amylase, the presence of MPO myeloperoxidase which is indicative of neutrophil infiltration, histological features and edema in some cases) was suppressed by the MA channel inhibitor GSMTx4 (Fig3) and required Piezo1 (at least in part). Acute pancreatitis was shown to be induced by the Piezo1 agonist Yoda1. Importantly, the effects of Yoda1 were partially dependent on Piezo1 expression in acinar cells, a critical finding that was surprisingly presented in supplemental information (Suppl. Fig 6), perhaps because of the variability observed (see below #3).

Response:

We appreciate the reviewer's suggestion regarding the study examining the effects of Yoda1 on pancreatitis in wild type and Piezo1^{aci} KO mice. These data were originally provided in supplementary Figure 6 which we have now moved into the main text as new Figure 5. As shown, the effects of Yoda1 on induction of pancreas are virtually eliminated in mice with acinar cell deletion of Piezo1 (Piezo1^{aci} KO mice). There was no statistically significant difference between Yoda1 treatment and control in Piezo1^{aci} KO mice.

Although the authors claim Piezo1 activates inflammatory pathways (Fig5) and cell death (Suppl Fig 4) by using (high) concentrations of Yoda1 applied to acinar cells in vitro, it is not clear that these processes could be initiated by activation of endogenous Piezo1 in an in vivo situation since they did not test Yoda1 effect's on KO animals (GsMTx4 is not specific for Piezo1).

Response:

To address the *in vivo* relevance of this pathway we did test Yoda1 in Piezo1^{aci} KO mice and found that the effects of Yoda1 were absent in Piezo1^{aci} KO mice. These data are shown in revised Figure 5. The *in vitro* effects of Piezo1 on acini from Piezo^{aci} KO mice are shown in revised Figure 2.

Nonetheless, this work sheds light on the molecular mechanisms that underlie pressure induced pancreatitis and reveals yet another organ system in which Piezos appear to play a role in pathophysiology.

Major concerns.

1. The in vivo pressure-induced pancreatitis model requires additional information for validation.

It is not apparent that the low mechano (5ul/min x 10 min= 50ul total infusion) is an adequate control for the high mechano (80uL/min x 5 min= 400uL total infusion) stimulation that produces pancreatitis-like symptoms.

Response:

In order to infuse at high pressure, it was necessary to increase both the volume of infusate and the rate of infusion. We used sham surgery as a control for the high pressure experiments. The low pressure condition did not produce biochemical or histological changes of pancreatitis. Therefore, for those experiments in which we tested the effects of Yoda1, which was administered into the pancreatic duct, we were able to infuse Yoda1 under conditions in which the infusion itself (low pressure) did not produce pancreatitis. This has been clarified in the manuscript.

a. The extent of infiltration of the dye in the high mechano condition (“pressure”) is not complete but it is hard to tell the extent from figure 1a. The reproducibility and extent of infiltration needs to be determined, especially since there is significant variability (see #3) that calls into question the conclusions. Is Fig1a taken immediately after the end of the injection or is it 24 hours after injection?

Response:

Each experiment was performed with methylene blue dye in the infusate so the extent of pancreas filling and any extravasation out of the pancreas could be detected immediately at the time of infusion. In the event that extravasation of infusate did occur (which was a rare event resulting from a technical mishap), the animal was immediately discarded from the study. The conditions we used in the current study design routinely filled the main pancreatic duct and the side branches throughout all regions of the pancreas including the head, body and tail of the gland but did not cause pancreatic duct disruption. The photograph shown in Figure 1 was taken immediately following infusion and dye can be seen throughout the gland. Of course, the intensity of the dye is most noticeable in the larger pancreatic ducts. Most of the infusate is absorbed within 24 hours of infusion and is no longer visible at the end of the experiment.

b. What is the extent of infusion into the pancreas during 5ul/min x 10min (“control”)? If the low condition only affecting the first 20-30% of the pancreas during the injection time, this calls into question how this could be a true “control” for the larger injection. It also begs the question:

where are the sections taken for histological analysis? This should be clearly stated and sections from different regions (proximal/distal to injection site; gastric, splenic, duodenal lobes) should be compared. The control sections need to be taken at the same location as experimental sections in comparison to the injection site. "Portions of the pancreata" is not specific enough. Where exactly were the sections taken? Are the sections consistent throughout the pancreas?

Response:

Infusate distributed throughout the head, body, and tail of the pancreas in both the low and high pressure conditions. Under high but not low pressure conditions, pancreatic inflammation was evident throughout the gland including the gastric, duodenal and splenic regions. To ensure consistency, we collected tissue from the mid region of the pancreas (corresponding to the body of the pancreas) for histological and biochemical assessments. These points have been further discussed in the revised manuscript.

2. Piezo1 underlies the acute pancreatitis observed by pressure in vivo. The critical figure 6 is problematic.

a. variability of control and pressure induced levels between figures 1,3 and 6 are cause for serious concern. Although the variability in basal conditions/fold increase does not appear to be due to strain background, it is not clear and should be considered and discussed. For instance, MPO control values in WT are ~4-fold higher in fig 6c compared to figs 1 and 3. The pressure induced MPO is at least 3-fold higher than that in figs 1 and 3.

Response:

We apologize for the confusion and thank the reviewer for allowing us to explain the differences. C57BL/6J mice were used in the experiments shown in Figures 1 and 3. The wild type mice shown in the original Figure 6 were Piezo1^{fl/fl} mice that were generated on a C57BL/6J background and are littermates of Piezo1^{aci} KO mice. The differences in these genotypes have been clarified in the revised text. Note the original Figure 6 is Figure 4 in the revised manuscript.

It is possible that the differences in genetic background (control C57BL/6J vs. wild type mice which were littermates of Piezo1^{aci} KO mice on a C57BL/6J background) could be responsible for the variability between experiments. Nevertheless, we observed a significant difference between the treated group that has a functional Piezo1 and the Piezo1^{aci} KO group. Another difference between experiments was that Piezo1^{aci} KO mice and their littermates were subjected to tamoxifen treatment while the C57BL/6J mice were not. Nevertheless, the finding that pharmacological or genetic deletion of Piezo1 in mice of slightly different backgrounds and experimental conditions exhibited similar patterns of protection against pancreatitis reinforces the concept that Piezo1 mediates pressure-induced pancreatitis.

b. The exact genotypes of the mice used in these studies is not clear. On p12 the fl/fl mice were defined as wildtype but then on p 13 "littermates were used for experiments". Are the control

“wildtype” mice really Cre-;Piezo1fl/fl mice? It should be clearly stated what genotype/background of mice were used in each experiment.

Response:

See response above. These points have been clarified in the revised manuscript.

c. what sections are compared for control and pressure?

Response:

The mid-region (body) of the pancreas was used for all histological and biochemical comparisons.

d. why was serum amylase not measured in these experiments?

Response:

We have found that serum amylase is an indicator of acute pancreatic injury but is not a reliable marker of severity of pancreatitis. Although it was included in this initial experiment, we do not believe it influences the interpretation of the results.

e. what do the n refer to in the bars in fig 6d? Does n= the number of sections from a single animal or number of animals from which a certain number of sections were scored? Or a combination? The number of separate animals and the number of sections from each animal needs to be clearly stated.

Response:

“n” refers to the number of animals in each group. This information has been added to the figure legend (new Figure 4).

3. Histopathology scoring. Since the GSMTx4 (fig3) and Piezo1aciKO (fig6) “reversal” of pressure induced damage is only partial, it is imperative to give values for each of the parameters scored. Is GsMTx4 or cKO partially reversing each of the parameters similarly or is GsMTx4 or cKO fully reversing some of the parameters (Piezo1 dependent) and having no effect on other parameters (Piezo1 Independent)? For instance, is edema and infiltration always dependent on Piezo1 but fat necrosis not Piezo1 dependent?

Histological parameters should be explicitly described and any differences from the 1989 paper should be discussed. The reference given refers to an earlier work by these authors in which the parameters are still not explicitly spelled out and refers the reader to yet another earlier, 1989 paper.

Response:

We have analyzed the histology scoring parameters separately and these data are now shown in Supplementary Figure 6. Both experiments indicate that all the parameters are reduced in a

similar pattern when either GsMTx4 is used in C57BL/6J mice or when KO mice are compared with their littermates. In this model of pancreatic injury, all of the pancreatitis parameters were affected in a similar manner.

4. In the abstract the authors claim: “to determine whether [the application of pressure within the gland] was through Piezo1 activation, we induced pancreatitis by intrapancreatic duct instillation of ... Yoda1.” To this reviewer, the Yoda1 experiments are weak. It is very concerning that Yoda1 infusion under low mechano stimulus appears to induce very variable effects in the pancreatitis parameters. Comparing Fig 4 with Suppl Fig6 suggests the observed variability may negate the significant effects shown in Fig. 4. For instance, with amylase: 10x increase observed in Fig 4b but only 2-3 fold in WT in Suppl Fig 6b. The cKO animals reveal a 2-3 fold increase in the presence of Yoda1. The baseline differences are substantial. The in vitro experiments on acinar cells indicate that there is a late (at >~1 min) increase in intracellular calcium by 25uM Yoda1 that is only partially piezo1 dependent. Taken together, the data indicate issues with variability and off-target effects of Yoda1.

Response:

Based on the reviewer’s recommendations, we have eliminated the experiment described in the original Figure 4. We have moved the original Supplementary Figure 6 into the body of the manuscript which is new Figure 5.

We agree with the reviewer that Yoda1 induced a rapid initial increase and a sustained rise in $[Ca^{2+}]_i$ beyond one minute of stimulation. Yoda1 application to HEK-293 cells with heterologous expression of Piezo1 induces an intracellular calcium rise due largely to Ca^{2+} influx and not Ca^{2+} release from intracellular stores (Syeda R et al, eLife 2015, DOI: 10.7554/eLife.07369). The rapid increase we observed in acinar cells following Yoda1 administration is consistent with direct effects on the Piezo1 ion channel. However, it is difficult to compare cells with endogenously expressed Piezo1 such as acinar cells with cells that express heterologous Piezo1 particularly since acinar cells have a plentiful source of stored intracellular Ca^{2+} that is highly regulated. At present, we can conclude that Yoda1 activates Piezo1 channels which induce an initial $[Ca^{2+}]_i$ rise. The sustained $[Ca^{2+}]_i$ rise is probably related to a calcium-activated calcium release pathway that follows Piezo1 activation. This possibility is the subject of future investigations.

a. What is the extent of the Yoda1 infusion throughout the pancreas? This is not disclosed.

Response:

Based on visual inspection of the gland following administration of solution containing methylene blue dye, Yoda1 administered under low pressure conditions appears to be distributed to at least 80% of the gland and always involved the body of the pancreas. For

tissue analyses, we took care to ensure that the mid-region of pancreas was used for all comparisons.

Concentration of Yoda1 used is very high; at 100uM, Yoda1 crashes out of aqueous solution. Although the Yoda1 concentration used in vitro is 25 uM (fig 2), it is not clear what Yoda1 concentration is used in Fig5B,C.

Response:

Figure 5 has now been eliminated and the LDH results have been included in revised Figure 2. We also appreciated that Yoda1 precipitates at high concentrations. We dissolved Yoda1 at 37°C and kept solution at this temperature before administration which partially reduced the problems with precipitation. Our intent was to induce a pathological condition to determine if Piezo1 activation could be linked to pancreatitis, however, it is possible that this may be an overestimation of the actual concentrations (Figure 2f and g).

b. The biggest concern is the use of a very high concentration of Yoda1 that likely would be activating more endogenous Piezo1 than would be activated by physiological pressure stimuli. Using the pressure model, are inflammatory mediators and mRNA changed in wildtype controls but not Piezo1cKO?

Response:

We do not know to what extent Piezo1 is activated under physiological conditions so it is difficult to know what concentration of Yoda1 fully recapitulates the in vivo condition. Our purpose was to determine whether Piezo1 alone could induce pancreatitis and the selectivity of Yoda1 for Piezo1 activation was used to address that question.

Once pancreatitis is induced there is a rapid influx of inflammatory cells with extensive cytokine production. We felt that it would not be possible to distinguish the direct effects of Piezo1 activation on acinar cytokine production from that produced by infiltrating inflammatory cells. Therefore, we did not measure pancreatic tissue cytokines in the in vivo pressure experiment.

c. Histology is incompletely reversed in cKO in pressure model, but is completely reversed in cKO in YODA1 induced acute pancreatitis. On the other hand, the reverse seems to be true for edema and MPO in Fig 6 (pressure model). Please discuss.

Response:

We do not have an explanation for the differences in responses between the experiments other than biological variability. In experiments using a fluorescent indicator for Cre expression, we do know that although extensive (>90%), Cre does not appear to be expressed in every acinar cell. Note the original Figure 6 is now Figure 4 in the revised manuscript.

d. Appropriate statistical analyses need to be provided for all relevant comparisons.

Response:

A symbol for a column signifies that the results for this column are significantly different than all the other conditions. In addition, the other conditions are not significantly different from each other.

Student's t test was performed when a comparison was made between two conditions. ANOVA with the Tukey post-test analysis was used in the rest of the figures. Different P values were designated by connecting lines between bars (e.g., Figure 2a). If one condition was significantly different with the same P value than all the other conditions, a P value symbol was only presented on the column representing this condition (see Figure 2e for example).

5. If the set of experiments used to conclude that Piezo1 activation induces inflammatory pathways is included in the manuscript it is imperative to show that the effects of Yoda1 on IκB and all the genes shown in 5B and 5C are dependent on Piezo1 activity using the cKO animals vs appropriate Cre- littermate controls. There appear to be issues with these experiments, for instance, Fig5a,b suggests reduction in IκB but there is no housekeeping (actin?) control for normalization.

Response:

We have taken the reviewer's advice and removed the cytokine data from the manuscript. The in vitro direct effects of Yoda1 on acinar cells from wild type and Piezo1^{aci} KO mice are now shown in Figure 2.

6. Statistical analyses are not included.

Fig5c: is NORMALIZED IκB reduction by 10uM yoda1 significant?

Response: Data removed

Fig 6d: what is the n referring to? How many sections were scored from how many different animals?

Response:

N refers to the number of animals per group. One section per animal was scored. The data in original Figure 6d are now Figure 4e in the revised manuscript.

Suppl Figure 4a: is 10uM yoda1 significantly increased compared to vehicle?

Response: There was no statistical difference.

Suppl Figure 4b: is 100uM yoda1 effect in presence of EGTA or GsMTx4 significantly different from EGTA or GsMTx4 controls?

Response: There was no statistical difference.

Suppl Figure 6b and 6c: in cKO animals, is Yoda1 (700uM) effect signif different from veh control?

Response: There was no statistical difference. This figure now is included in the main manuscript.

Minor.

Fig1a. what is the concentration of methylene blue used?

Response: 1% methylene blue was used in the infusate solution.

Fig 1: why wasn't edema measured in these experiments?

Response: We have added these data to the figure 1.

Fig2e. why is DMSO applied when Yoda1 is added if this is the vehicle for Yoda1?

Response: DMSO is the vehicle for Yoda1 and was used as a vehicle control. The figure legend has been clarified.

Fig6. Include plasma amylase results for pressure model using Piezo1 cKO animals.

Response: Same as number point 2d discussed above. We did not see a significant difference in serum amylase levels. However, we have observed that serum amylase is not a good indicator of pancreatitis severity and does not correlate with other pancreatitis parameters. Note Figure 6 in the original manuscript is now Figure 4 in the revised manuscript.

Suppl fig 4a: Was the LDH release from isolated acini at 10uM Yoda1 significantly different from vehicle?

Response: No, there was no significant difference.

Methods p. 13: EYFP;Piezo1aciKO are stated to express truncated Piezo1 in acinar cells. Is this predicted or tested experimentally?

Response:

In this particular case, this was predicted. However, in each experiment we confirmed that genetic recombination had occurred. Every KO mouse that was used in our experiments was genotyped by PCR in order to detect a specific DNA fragment that is visible only after ablation of the 3 exons from the Piezo1 gene.

Reviewer #3 (Remarks to the Author):

This is a highly novel and important finding with significant clinical implications.

Liddle et al have followed up on the recent discovery of Piezo-1 and Piezo-2, which are pressure activated ion channels in the etiopathogenesis of acute pancreatitis.

They have demonstrated convincingly that Piezo-1 is expressed in pancreatic acinar cells and that pathophysiological Piezo1 activation is associated with pressure-induced acute pancreatitis, whilst blocking of Piezo1 channels and targeted Piezo-1 deletion abrogated the effects of pressure in the induction of acute pancreatitis.

They could have gone on to show that acute pancreatitis could still be induced by other trigger factors such as bile acids and/or fatty acid ethyl esters in Piezo-1 deleted/blocked acini... additive? synergistic? etc.

Response:

We thank the reviewer for the positive comments and suggestions. Our initial goal was to determine the mechanism for the clinical observation of pressure-induced injury to the pancreas but we would very much like to follow-up the current investigation with studies to determine the relative contribution of pressure to other modes of pancreatitis including bile acid-induced pancreatitis.

Certainly this should be picked up in the discussion. In gallstone pancreatitis there is always the passage of multiple gallstones into the duodenum and this will inevitably result in both the temporary obstruction to both the main bile duct and main pancreatic duct and then bile reflux into the main pancreatic either directly or from the duodenal lumen. Do bile acids promote acute pancreatitis under low pressure conditions (ie just above normal pressure) ?

Acosta JM, Ledesma CL. Gallstone migration as a cause of acute pancreatitis. N Engl J Med. 1974;290(9):484-487;

Neoptolemos JP et al. Results of a prospective randomized trial of ERCP and endoscopic sphincterotomy in acute gallstone pancreatitis. Lancet 1988; ii: 979-983;

van Santvoort HC et al. Early endoscopic retrograde cholangiopancreatography in predicted severe acute biliary pancreatitis: a prospective multicenter study. Ann Surg. 2009;250(1):68-75.

Response:

Although at the time, we did not measure the actual pressure of infusion, we believe that under certain circumstances bile acids may induce pancreatitis even at low pressure (Shahid R, et al. Cell Mol Gastroenterol Hepatol 2015, 1:75-86). We completely agree with the reviewer that it will be necessary to determine the extent pressure affects the initiation and progression of pancreatitis in this and other common clinical conditions. Thank you for the references.

Thus the therapeutic importance of blocking Piezo-1 would be considerable in acute pancreatitis.

Moreover there needs to be some discussion about chronic pancreatitis – hereditary, idiopathic, and other (alcohol etc). This is a progressive disease promoted not only ongoing trigger factors (genetic mutations, smoking etc) but also by gross and micro degrees of obstruction.

Response:

Thank you for the suggestion. We have expanded the discussion to include these important clinical points.

This study would promote the importance of free drainage of major ducts but also of Piezo-1 blockade to lessen the impact of intermittent small duct obstruction.

Finally in the Introduction they mention that abnormally high intracellular calcium concentrations lead to impaired zymogen granule processing and fusion of zymogen granules with lysosomes causing trypsinogen activation.

But Raraty et al showed that trypsin activation and vacuolization are caused directly by the rise in intracellular calcium concentration without the requirement for co-localization.

Ref 17. Raraty, M. et al. Calcium-dependent enzyme activation and vacuole formation in the apical granular region of pancreatic acinar cells. Proc Natl Acad Sci U S A 97, 13126-13131 (2000).

Response:

Thank you for pointing this out. We have modified the text and added these references.

Reviewer: John Neoptolemos

REVIEWERS' COMMENTS:

Reviewer #2 (Remarks to the Author):

We are happy with this version of the Romac, Shahid et al., (NCOMMS-17-19425) manuscript on Piezo1 in pancreatitis.

1) The authors show convincingly using multiple in vitro and in vivo methods and parameters that Piezo1 mediates acute pancreatitis. There is only issue which should not detract from the paper being published (because many other parameters indicative of pancreatitis are shown): histological parameters are scored from only 1 section per mouse. The authors need to clearly acknowledge this in the methods. For example, on p. 15, end of last line: ADD SOMETHING LIKE: "Histological studies were performed on one section taken from the central body of the pancreas per mouse for the number of mice indicated. Criteria used to chose the section from the central part of the pancreas are x, y, z").

2) 3 recent papers indicate that mPiezo1 has more than 14 TM domains; p4 line 2 should be revised to indicate the latest number and the latest papers should be cited (MacKinnon; Patapoutian; Xiao labs).

Reviewer #3 (Remarks to the Author):

Good response to reviewers.

Excellent contribution to the field.

John Neoptolemos.

Response to Reviewer Comments

Reviewer #2 (Remarks to the Author):

1) The authors show convincingly using multiple in vitro and in vivo methods and parameters that Piezo1 mediates acute pancreatitis. There is only issue which should not detract from the paper being published (because many other parameters indicative of pancreatitis are shown): histological parameters are scored from only 1 section per mouse. The authors need to clearly acknowledge this in the methods. For example, on p. 15, end of last line: ADD SOMETHING LIKE: "Histological studies were performed on one section taken from the central body of the pancreas per mouse for the number of mice indicated. Criteria used to choose the section from the central part of the pancreas are x, y, z").

Thank you for mentioning this issue. We have addressed this by showing the section of the pancreas that was used for histological analyses (Fig. 1b, yellow outline). After resection in toto, a 15-20 mg fragment of tissue was taken from the mid-portion of the pancreas. Out of this fragment a section with an area of $\sim 25 \text{ mm}^2$ was used for embedding and was then processed of H and E staining.

2) 3 recent papers indicate that mPiezo1 has more than 14 TM domains; p4 line 2 should be revised to indicate the latest number and the latest papers should be cited (MacKinnon; Patapoutian; Xiao labs).

Thank you for indicating the latest information of Piezo1 structure; these references have been added.